

# Trends in mortality rates of cutaneous melanoma in East Asian populations

Ling Chen[1,2] and  Shaofei Jin[3]

[1] Zhongshan School of Medicine, Sun Yat-sen University, Guangzhou, China
[2] Immunotherapy Institutes, Fujian Medical University, Fuzhou, China
[3] Northeast Institute of Geography and Agroecology, Chinese Academy of Sciences, Changchun, China

## ABSTRACT

The incidence of cutaneous melanoma (CM) has rapidly increased over the past four decades. CM is often overlooked in East Asian populations due to its low incidence, despite East Asia making up 22% of the world's population. Since the 1990s, Caucasian populations have seen a plateau in CM mortality rates; however, there is little data investigating the mortality rates of CM in East Asian populations. In this study, the World Health Organization Mortality Database with the joinpoint regression method, and a generalized additive model were used to investigate trends in age standardized mortality rates (ASMRs) of CM in four East Asia regions (Japan, Republic of Korea (Korea), China: Hong Kong (Hong Kong), and Singapore) over the past six decades. In addition, mortality rate ratios by different variables (i.e., sex, age group, and region) were analyzed. Our results showed ASMRs of CM in East Asia significantly increased non-linearly over the past six decades. The joinpoint regression method indicated women had greater annual percentage changes than men in Japan, Korea, and Hong Kong. Men had significantly greater mortality rate ratio (1.51, 95% CI [1.48–1.54]) than women. Mortality rate ratios in 30−59 and 60+ years were significant greater than in the 0−29 years. Compared to Hong Kong, mortality rate ratio was 0.72 (95% CI [0.70–0.74]) times, 0.73 (95% CI [0.70–0.75]) times, and 1.02 (95% CI [1.00–1.05]) times greater in Japan, Korea, and Singapore, respectively. Although there is limited research investigating CM mortality rates in East Asia, results from the present study indicate that there is a significant growth in the ASMRs of CM in East Asian populations, highlighting a need to raise awareness of CM in the general population.

## INTRODUCTION

Over the past four decades, cutaneous melanoma (CM) has been one of the most rapidly increasing cancers globally (*Garbe & Leiter, 2009*; *Giblin & Thomas, 2007*; *Nikolaou & Stratigos, 2014*). Numerous studies investigated the incidence and mortality rates of CM in Caucasian populations (*Barbaric et al., 2016*; *Wallingford et al., 2013*; *Bristow et al., 2013*; *Duschek et al., 2013*; *Sneyd & Cox, 2013*), however, only a small number of studies exist focusing on the incidence and mortality rates of CM in East Asian populations. Research investigating average annual age-standardized incidence rate of CM in Japan was 0.25 per 100,000 for men, and 0.20 per 100,000 for women between 1964 and 1995 (*Tanaka et al.,*

Corresponding authors
Ling Chen, chenlsysu@hotmail.com
Shaofei Jin, jinsf@tea.ac.cn

*1999*). In China: Hong Kong (Hong Kong), the mean incidence of CM between 1983 and 2002 was 0.8 per 100,000 for men and 0.6 per 100,000 for women, but the mortality rate significantly increased during this time (*Makredes, Hui & Kimball, 2010*). In Singapore, the annual age-standardized incidence of all skin cancer increased between 1968 and 1997 (*Koh et al., 2003*) and in Beijing and Shanghai, the mortality of CM increased between 1988 and 2007 (*Zeng et al., 2012*).

Although mortality rates of CM in East Asia are relatively low, East Asians constitute 22% of the world's population (*United Nations, 2015*) therefore, identifying trends in mortality and the characteristics of this disease in this population is imperative. Risk factors affecting mortality rates of CM include ultraviolet (UV) radiation exposure (*Nikolaou & Stratigos, 2014*), which is highly variable in this geographically spread region. Aging is another risk factor affecting mortality rates of CM (*Simard et al., 2012*). East Asia, like the rest of the world, has a rapidly aging population, which has serious implications for sustainable social development (*Goh, 2005*; *Oizumi, 2013*). In 2015, 26% of population in Japan and 13% of population in Republic of Korea (Korea) were aged 65 or older (*World Bank, 2016*). Finally, there are sex-related differences in CM mortality rates (*Lasithiotakis et al., 2008*) with men being at a greater risk of developing and dying from CM (*Luk et al., 2004*; *Makredes, Hui & Kimball, 2010*). In Hong Kong, the female-to-male incidence rate is 1:1.22 (*Luk et al., 2004*), and the mortality rate is 1:1.35 (*Makredes, Hui & Kimball, 2010*). Few studies exist investigating the cumulative effects of risk factors on the mortality rate of CM in East Asian populations; this study aims to address this gap.

To address this research gap, long-term trends in CM mortality in four regions including: Japan, Korea, Hong Kong, and Singapore, were analyzed using the joinpoint regression method and Generalized Additive Model (GAM) over the past six decades. The specific aims of this study are to identify trends in mortality rates of CM and the mortality rate ratio of CM by sex and age in four specific East Asian regions over the past six decades.

## METHODS AND MATERIALS
### Data source
Mortality data from East Asia were obtained from the World Health Organization Cancer Mortality Database (*World Health Organization, 2016*). This database is managed by the Section of Cancer Surveillance at the International Agency for Research on Cancer. Age standardized mortality rates (ASMRs) and deaths caused by CM in four East Asia regions were extracted over different periods: Japan (1955–2013), Korea (1985–2013), Hong Kong (1966–2013), and Singapore (1968–2014). The records were defined according to the International Classification of Diseases (ICD) codes as ICD-8 172, ICD-9 172 and ICD-10 C43. The ASMRs data were imported directly from the database and registered by sex and age. ASMRs are calculated as a weighted mean based on the world standard population in age-specific rates (*Segi, 1960*),  and are defined as the total number of CM deaths per 100,000 persons. The data were further categorized into three age groups: 0–29 years, 30–59 years, and 60+ years. See Supplemental Information 1 for the detailed dataset.

## Statistical analysis

The joinpoint regression method and Generalized Additive Model (GAM) were performed to investigate the trends of ASMRs in four East Asian regions over the past six decades. The joinpoint regression method detects change points for trend data (i.e., mortality rates of cancer). This method starts with a minimum number of joinpoint (default, 0), and adds more jointpoints to the models if more statistically significant linear changes ($P < 0.05$) are found after testing using a Monte Carlo Permutation method. To avoid the occurrences of spurious changes in trends, the maximum jointpoints were set to three (*Baade et al., 2012*). Finally, the model will return the estimated parameters and their statistical power values among the joinpoints. All joinpoint analysis was conducted using joinpoint regression software (Version 4.3.10—April 19, 2016), downloaded from the Surveillance Research Program of the US National Cancer Institute (http://surveillance.cancer.gov/joinpoint/). For further details on this method, see *Kim et al. (2000)*. In the present study, this method is used to analyze average annual percentage changes (AAPC) and annual percentage changes (APC) of ASMRs between men and women for each region.

GAM was applied to investigate the relationship between ASMRs of CM and other factors, including gender, age group, year, and region. GAM is a more flexible statistical model to analyze nonlinear relationships between variables. GAMs were conducted using the *mgcv* package of R programming (*R Core Team, 2016*). Assuming the deaths follow a Poisson distribution, the GAM models were expressed as follows:

For the four regions, the model was described as follows (Eq. (1)):

$$\text{Log}(\text{ASMRs}_i) = a + s(\text{year}_i) + \text{factor}(\text{agegroup}_i) + \text{factor}(\text{sex}_i)$$
$$+ \text{factor}(\text{region}_i) + \varepsilon_i \varepsilon_i \sim N(0, \sigma^2). \tag{1}$$

For each region, the model was described as follows (Eq. (2))

$$\text{Log}(\text{ASMRs}_i) = a + s(\text{year}_i) + \text{factor}(\text{agegroup}_i) + \text{factor}(\text{sex}_i) + \varepsilon_i \varepsilon_i \sim N(0, \sigma^2) \tag{2}$$

where, ASMRs refers to age standardized mortality rate of the year $i$; year refers to the year; age group, sex, and region refers to nominal explanatory variables of different age groups (three levels), gender (two levels: male and female), and the four regions (four levels), respectively. $a$ refers to the intercept; E$i$ refers to the error term. The $s(\text{year}_i)$ is the smoothing function to show the trends in AMSRs over the periods. See *Wood (2006)* for technical detail of GAM implementing in R, and Fig. S9 for simplified explanation.

## Mortality rate ratios of cutaneous melanoma by different explanatory variables

Mortality rate ratios of CM by sex, age group, and regions are obtained from estimated parameters of explanatory variables in the GAMs, and express the relative contribution of different explanatory variables to the ASMRs. The mortality rate ratios are set to references for the factor of female, and 30–59 years age group, and Hong Kong, respectively.

## RESULTS

### Annual percentage changes of cutaneous melanoma in East Asia

Table 1 showed the average ASMRs for male was greater than for female in each region over past decades. Moreover, the average ASMRs increased from the northernmost region to the southernmost region for both genders. Female conducted greater AAPC than male in the four regions except Singapore. ASMRs increased significantly for both sexes in Japan and Korea over the entire periods (Table 1). For details, ASMR was increasing continuously since 1985 by 7.4% (95% CI [5.6%–9.2%]) yearly in Korea female (Table 1). Further, although more than one joinpoints were detected for both Japanese and Korea male, all APCs in the last trends were significant greater than zero. For the other two regions, only significantly growth in ASMRs was found in female of Hong Kong. For the regional differences in APCs, Korea has the greatest APCs among the four regions, while no changes were found in Singapore. Furthermore, significant increases in the ASMRs were found over the last five years in all regions except Singapore. Displays of joinpoint regression analyses for male and female in the four regions were shown in Figs. S1–S8 (Supplemental Information 2). In addition, the GAM showed the overall ASMRs of East Asia population had a significant non-linear growth over the past six decades (Fig. S9).

### Mortality rate ratios of cutaneous melanoma by sex, age group, and region in East Asia

Significant differences in mortality rate ratio of different explanatory factors were found among the levels (Figs. 1A–1C). For the factor of sex, men had significantly higher mortality rate ratio than women (1.51; 95% CI [1.48–1.54]; $P < 0.001$, Chi.sq = 1,970, Chi-square test, same with below, Fig. 1A). For the factor of age group, mortality rate ratio were significantly greater for the 30–59 age group and 60+ age group compared to the 0−29 age group (Fig. 1B, $P < 0.001$, Chi.sq = 25,061). Finally, significant differences in mortality rate ratio were detected among the four regions (Fig. 1C, $P < 0.001$, Chi.sq = 1,254). Mortality rate ratio in Japan (0.72; 95% CI [0.70–0.74]), Korea (0.73; 95% CI [0.70–0.75]), and Singapore (1.02; 95% CI: [1.00–1.05]) were significantly different with Hong Kong.

In addition, we analyzed the specific mortality rate ratios by sex and age group in different regions due to their geographic locations. Table 2 showed men had significantly greater mortality rate ratios than women (95% CIs > 1). Among the four regions, the greatest mortality rate ratio was found in Hong Kong (1.57; 95% CI [1.52–1.62]), while the lowest was in Singapore (1.43; 95% CI [1.38–1.47]) despite their southerly geographic locations. Further, mortality rate ratio in Singapore (1.43, 95% CI [1.38–1.47]) was significantly less than in Japan (1.54; 95% CI [1.48–1.59]) and in Hong Kong (1.57; 95% CI [1.52–1.62]). Moreover, although East Asia spreads geographically, Table 2 showed no significant changes in mortality rate ratios in 60+ age groups were found among the four regions.

## DISCUSSION

In the present study, trends in mortality rates of CM in four East Asian regions were analyzed using the GAM and the joinpoint regression method. Main general findings

Chen and Jin (2016), *PeerJ*, DOI 10.7717/peerj.2809

**Table 1** Joinpoint regression analysis of age-standardized mortality rate of cutaneous melanoma in four Asia regions.

| Region[a] | Sex | Period | Average ASMR[b] | AAPC[c] (95% CI)[d] | Trend 1 | | Trend 2 | | Trend 3 | | AAPC (95% CI)[f] |
|---|---|---|---|---|---|---|---|---|---|---|---|
| | | | | | Years | APC[e] (95% CI) | Years | APC (95% CI) | Years | APC (95% CI) | |
| Japan | Male | 1955–2013 | 0.18 | **2.5 (2.1, 2.8)**[g] | 1955–1973 | **7.4 (6.5, 8.4)** | 1973–2013 | **0.3 (0.0, 0.6)** | | | **0.3 (0.0, 0.6)** |
| Japan | Female | 1955–2013 | 0.12 | **2.6 (2.1, 3.2)** | 1955–1974 | **6.7 (5.1, 8.4)** | 1974–2013 | **0.7 (0.4, 0.9)** | | | **0.7 (0.4, 0.9)** |
| Korea | Male | 1985–2013 | 0.23 | **7.3 (0.7, 14.4)** | 1985–1989 | 0.1 (−16.6, 20.1) | 1989–1992 | 43.7 (−19.4, 156.0) | 1992–2013 | **4.3 (2.8, 5.9)** | **4.3 (2.8, 5.9)** |
| Korea | Female | 1985–2013 | 0.16 | **7.4 (5.6, 9.2)** | 1985–2013 | **7.4 (5.6, 9.2)** | | | | | **7.4 (5.6, 9.2)** |
| Hong Kong | Male | 1966–2013 | 0.28 | −0.5 (−3.8, 2.9) | 1966–1969 | −28.4 (−57.6, 20.9) | 1969–2013 | **1.8 (0.9, 2.7)** | | | **1.8 (0.9, 2.7)** |
| Hong Kong | Female | 1966–2013 | 0.2 | **2.4 (1.4, 3.4)** | 1966–2013 | **2.4 (1.4, 3.4)** | | | | | **2.4 (1.4, 3.4)** |
| Singapore | Male | 1968–2014 | 0.31 | −0.4 (−1.7, 0.9) | 1968–2014 | −0.4 (−1.7, 0.9) | | | | | −0.4 (−1.7, 0.9) |
| Singapore | Female | 1968–2014 | 0.23 | −0.2 (−1.5, 1.0) | 1968–2014 | −0.2 (−1.5, 1.0) | | | | | −0.2 (−1.5, 1.0) |

**Notes.**
[a] Korea: Republic of Korea. Hong Kong: China: Hong Kong.
[b] ASMR: Age standardized mortality rate, per 100,000 persons.
[c] AAPC: Average annual percentage changes over the entire period.
[d] 95% CI: 95% confidence interval.
[e] APC: Annual percentage changes.
[f] AAPC over the last five years.
[g] Bold values represents joinpoint is significantly different from zero at $P < 0.05$ level.
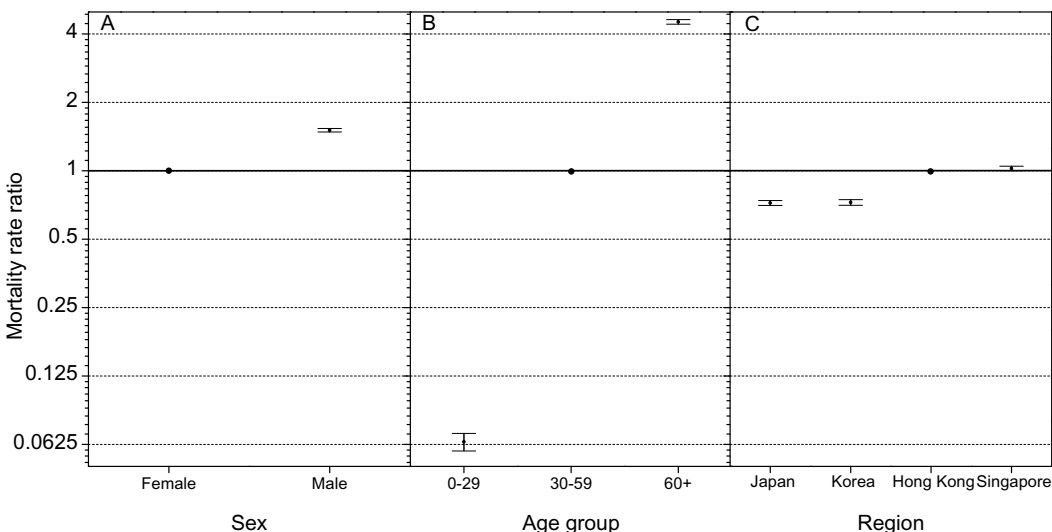

**Figure 1  Mortality rate ratios of cutaneous melanoma by (A) sex, (B) age group, and (C) region in East Asia.** The error bars represent 95% confidence intervals. 0–29: deaths aged 0–29 years; 30–59: deaths aged 30–59 years; 60+: deaths aged 60+ years. The mortality rate ratios of male, 30–59 years, and Hong Kong are set to reference group.

**Table 2  Mortality rate ratio (with 95% confidence intervals) of cutaneous melanoma by sex and age group in different regions.**

|  | Japan | Korea | Hong Kong | Singapore |
|---|---|---|---|---|
| **Sex** | | | | |
| Female (reference) | 1.00 | 1.00 | 1.00 | 1.00 |
| Male | *1.54 (1.48, 1.59) | *1.52 (1.45, 1.60) | *1.57 (1.52, 1.62) | *1.43 (1.38, 1.47) |
|  | $p < 0.001$ | $p < 0.001$ | $p < 0.001$ | $p < 0.001$ |
| **Age group (years)** | | | | |
| 0–29 | *0.08 (0.07,0.09) | *0.07 (0.05,0.08) | *0.06 (0.05,0.07) | *0.06 (0.05,0.07) |
| 30–59 (reference) | 1.00 | 1.00 | 1.00 | 1.00 |
| 60+ | *4.39 (4.18,4.60) | *4.44 (4.18,4.71) | *4.75 (4.55,4.96) | *4.46 (4.28,4.65) |
|  | $p < 0.001$ | $p < 0.001$ | $p < 0.001$ | $p < 0.001$ |

**Notes.**
*$p < 0.001$.

include: (1) significant growth in ASMRs in four East Asian regions; (2) mortality rate ratios were significantly greater in men compared to women in East Asian populations; and (3) mortality rate ratios were significantly greater in groups aged 60+ years compared to those aged 30–59 and 0–29 years in East Asian populations.

## Link with international studies

Over the past several decades, considerable studies investigated mortality rates of CM in Caucasian populations due to the dramatic growth in incidence and mortality (*Garbe & Leiter, 2009*). Conversely, trends in incidence and mortality in Asian populations have been overlooked. The present study aims to fill gaps and expand the discussion by investigating trends in mortality rates and the mortality rate ratios by gender and age.

Mortality rates of CM in some European countries and North America had leveled off since the 1990s; however, the results of this study suggested this may not be true based on recent researches for Caucasian populations and this study for Asian populations. In the present study, ASMRs increased significantly in four regions of East Asia over the past six decades, with female contributing greater AAPC than male in Japan, Korea, and Hong Kong. In the other region worldwide, although mortality rate of CM leveled off in several regions, e.g., Spain (*Cayuela et al., 2005*) and Sweden between 1980s and 1997 (*Cohn-Cedermark et al., 2000*), more recent studies found the mortality rates of CM increased worldwide, e.g., USA (*Garbe & Leiter, 2009*), Nordic counties (*Tryggvadottir et al., 2010*), Brazil (*Mendes, Koifman & Koifman, 2010*), and Australia (*Baade & Coory, 2005*). For the age effects, older patients (>65 years old) had significant growth in mortality rates in some regions, e.g., South-Eastern European (*Barbaric et al., 2016*), Netherland (*Hollestein et al., 2012*), USA (*Simard et al., 2012*), and Australia (*Baade & Coory, 2005*). The updated trends in mortality rates of CM varied in different regions globally. It is worth to note more regions had reported increases in mortality rates after the leveled off of mortality rates of CM since 1990s (*Nikolaou & Stratigos, 2014*). These latest studies challenged the efforts and our knowledge of improving survival rates of CM through effective protection and identifying risk factors.

Furthermore, the results of the present study showed mortality rate ratio of men were significantly greater than that of women in four regions in East Asia. Results were consistent with mortality rates reported in men and women in Caucasian populations (*Bristow et al., 2013*). Further, results indicated that older individuals had the greatest risk of mortality from CM in four East Asian regions. This is consistent with previous conclusion in other regions. A recently published study indicated a significant growth of mortality in Southern Europe in middle and older aged individuals (*Barbaric et al., 2016*). In USA, there has a high incidence of CM for patients older than 65 years (*Simard et al., 2012*). In New Zealand, *Liang, Robinson & Martin (2010)* showed the older men (>59 years) had the highest risk of developing CM (*Liang, Robinson & Martin, 2010*).

UV radiation is as a key environmental risk factor for CM. The UV index, developed through an international effort, is a measure of the UV radiation on a scale from one to values greater than 11. Higher UV index values indicate a greater risk of skin damage; UV index value greater than two indicates needs for protection (*World Health Organization, 2002*). Due to the vast geographic spread of the four regions in this study, UV indexes varied, with higher UV index reported closer to the equator. Therefore, there was a UV index gradient from the southernmost region to the northernmost region in East Asia. In Singapore, the southernmost region (1°N), the maximum UV index ranged from 10 to 13 (Ultraviolet radiation and the INTERSUN Programme; http://www.who.int/uv/intersunprogramme/activities/uv_index/en/index3.html (accessed 4 July 2016)). In Hong Kong (22°N), the reported maximum UV index was greater than six in 69.8% of days over the past decade (Hong Kong Observatory; http://gb.weather.gov.hk/wxinfo/uvindex/chinese/cfreq_dist.htm (accessed 4 July 2016)). The maximum UV index in Japan (using Tokyo as the location, 36°N) ranged from 5 to 10 between March and September (Ultraviolet radiation and the INTERSUN Programme;

http://www.who.int/uv/intersunprogramme/activities/uv_index/en/index3.html (accessed 4 July 2016)). Thus, in East Asia, a UV index gradient exists among the four East Asia regions from Japan to Singapore. Moreover, the average ASMRs increased from the northernmost region to the southernmost region for both genders. This suggests the geographic gradient may play a role in relative risks in East Asia. This result was consistent with studies in other regions e.g., the Europe (*Crocetti et al., 2015*). UV is considered one of the main environmental risk factor to developing CM, therefore, sun protection is as an effective way to reduce the risk of damaging skin (*Guy et al., 2015*).

## Limitations

There are several possible improvements of our study. Firstly, given the age distribution from the previous studies, we did not use the standard 5-year age group division to examine the trends in ASMRs of East Asia. *Sng et al. (2009)* reported CM incident rates in Singapore in two age groups: younger than 60 years, and older than 60 years. *Luk et al. (2004)* reported that 57.1% of CM patients in Hong Kong received a diagnosis at age 60 or greater, while only 6.3% received a diagnosis at age 30 or younger, and no data was reported for patients between the age of 50 and 60 years old. In addition, *Hui et al. (2005)* reported a mean age of 57.6 years for 32 Hong Kong CM patients. Given these information, ASMRs data was categorized into three age groups to allow for comparison to previous work. Thus, more will be carried out for more detail age groups.

Secondly, the coverage of data is limited for East Asia. Due to the lack of mortality data in other regions of East Asia, ASMRs were only analyzed for Japan, Korea, Hong Kong, and Singapore in this study. Although China has the largest population in the world, no long-term data were available recently. Furthermore, it is important to note that all four regions are high-income. The Gross Domestic Production per capita in Japan, Korea, Hong Kong, and Singapore was $36,194.40, $27,970.50, $40,169.50, and $56,284.30, respectively. Therefore, more data of CM in the developing region are needed in the future.

## Public health implication

Due to low incidence rates of CM in Asians populations, data on this topic were limited (*Bellew, Del Rosso & Kim, 2009*). This study demonstrates there was more risk for mortality in men, older aged individuals (60+), and populations with high UV indexes in four East Asian regions. In order to reduce the incidence and mortality of CM, for the public, it is suggested to: (1) reduce UV radiation exposure; (2) increase sun protection; (3) avoid prevalent sunbed use (*Wallingford et al., 2013*); (4) increase public awareness/education (*Buster et al., 2012*; *Gohara, 2015*); and (5) adopt screening and early diagnosis practices (*Boniol, Autier & Gandini, 2015*; *Crocetti et al., 2015*; *Tryggvadottir et al., 2010*).

Although the CM mortality data in East Asian populations is limited, the significant increase in mortality rates identified in this study point to a need for increased attention and further study. Finally, it is vital to increase public awareness about the risks of CM and take action to reduce the incidence of CM.

## ACKNOWLEDGEMENTS

We acknowledge the excellent work of the Section of Cancer Surveillance, International Agency for Research on Cancer. We thank Prof. Peter Baade and the two anonymous reviewers for their constructive comments that greatly enhanced this manuscript. We would also like to thank Laura Beamish at the University of British Columbia for her assistance with English language and grammatical editing of the manuscript, and Prof. Jian Shuai at the Jiangxi University of Traditional Chinese Medicine for his assistance with statistical analysis.

### Funding

This work is supported by funds from Fujian Province Department of Science and Technology Research Program (2014Y2001 and 2014Y4008). The funders had no role in study design, data collection and analysis, decision to publish, or preparation of the manuscript.

### Grant Disclosures

The following grant information was disclosed by the authors:
Fujian Province Department of Science and Technology Research Program: 2014Y2001, 2014Y4008.

### Competing Interests

The authors declare there are no competing interests.

### Author Contributions

- Ling Chen and Shaofei Jin conceived and designed the experiments, performed the experiments, analyzed the data, contributed reagents/materials/analysis tools, wrote the paper, prepared figures and/or tables, reviewed drafts of the paper.

### Data Availability

The raw data has been supplied as Supplementary Files.

### Supplemental Information

Supplemental information for this article can be found online at http://dx.doi.org/10.7717/peerj.2809#supplemental-information.

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
