# Peer review of "Trends in mortality rates of cutaneous melanoma in East Asian populations"

_PeerJ, doi:10.7717/peerj.2809_

## Round 0.1 · original submission · Major Revisions

· Academic Editor

Major Revisions

Thank you for your submission. Please ensure that you consider each of the reviewer's comments and clearly outline how (and where) you have addressed each one in the revised manuscript.

Please accept my apologies for the time taken to provide these comments to you. It look a lot longer than expected to find suitable reviewers.

Reviewer 1 ·

Basic reporting

Generally fine as a descriptive epideiological manuscript.

Experimental design

CM is not common cancer in east Asia and the risk factor is relatively clear, so that this cancer doesn’t have special characteristics and short of importance in public health.

Validity of the findings

might be simple to compare the mortality rates among the four areas; analyze APCs. The relative effect is not necessay: mortality increases by age, male higher than female, just like most cancers that don’t need to test. Importantly, Hongkong is just a region of China, not a country (marked as country in some places).

Additional comments

This manuscript described CM mortality in 4 areas using cancer mortality database from WHO. The quality of this manuscript is not good and have a room of improvement in term of writing skill, analysis and explanation of results. Some minor errors below:
1. Line 48,In Singapore, the annual age-standardized incidence of all skin cancer increased between 1968 and 1997. It doesn’t mean CM increased.
2. Line 52, mortality rate reaching 0.10 per 100000 and 0.22 per 100000 in 2007, in several regions including Beijing and Shanghai respectively. Hard to understand.
3. Line 56, Although mortality---cancer mortality?
4. Line 74, mortality---CM mortality
5. World Health Organization Cancer Mortality Databaseis managed by IARC. As I know, it is not correct. Please check.
6. 1996 world standard population in age-specific rates (Segi 1969)?
7. old age group (60 - 85+ years old)---60+?
8. Which year UV index were used? UV might change by year?
9. I wonder why the CI varied in a so wider range in Korea and Hongkong female? Might be wrong.

Reviewer 2 ·

Basic reporting

The paper appears to adhere to PeerJ policies.
The English should be improved to bring it into line with standards of scientific reporting and expectations of the Peer J. Most of this is minor, I have not attempted to correct this draft, but could be rectified by an editor. One example, however, is the use of descriptors for the age groups chosen, “young” “middle” and “old” is too subjective. 60 year olds may not relate to being “old”. You could simply use the age ranges themselves, no need to give them terms. A second example is the way in which the authors use the term for the reference category “young age group was referred to one”. Better to say “ XX age group was used as the reference group”.
The description of trends in other countries, beginning at about line 308 is mechanical and repetitive. The authors could summarise these trends in a more reader friendly manner.
The introduction and background place this work in context. It appears to fill the gap in describing for the first time death rates from cutaneous melanoma in four selected East Asian populations. Other international reports and descriptions of trends in melanoma rates are given in the reference section.
This is an observational study and the authors have broadly used the STROBE reporting structure within the manuscript. There are however some departures from this general structure that should be addressed, eg:
1. Justification of the age distributions used with references to other papers is good practice (line 94 onwards), but there is too much detail in the methods. It would be preferable to state which age groups were chosen in the methods and then provide justification in the discussion.
2. The methods section contain details that would be more appropriate in the discussion. Eg, The section on “Brief introductions of the four East Asia regions” contains descriptive details about the regions and then the UV index for those regions that should be in the discussion section.
Charts and figures are generally clear and descriptive. Use of titles could be improved, eg Figure 4 instead of using single letters “J” “K” “H”, “S” it would be better to use.

Experimental design

The work represents original work, albeit reproduction and value-adding data available from the WHO Cancer Mortality Database. The authors have used Joinpoint Analysis and Generalized Additive Models to describe the trends in mortality in 4 East Asian countries.
The presentation of the “relative effects on ASMRs” throughout the paper is a slightly unconventional way of describing the differences in rates between groups. A brief guide to interpretation of these outcomes would be useful.
The title and abstract make this aim clear and the paper answers the questions posed.
The methods are described reasonably clearly (see my comments about GAMs). As the data are publically available no ethics approvals were necessary.
The inclusion of data on UV levels in each jurisdiction appears to be a major weakness in this paper due to the highly ecological nature of this type of analysis. Figure 4 provides a correlation between mortality and UV levels. This was not stated as an aim and appears to be an afterthought. I would avoid formal analysis of this type and rather use the discussion to propose this as a possible related factor.
There is no mention, aside from the final section of the discussion about UV prevention programs or about changes in diagnosis or treatment of CM or about subtle genetic differences that may predispose populations to death from CM. Brief discussion of how each may impact death rates in the areas chosen for analysis would be valuable.

Validity of the findings

The data are robust population wide collections available from WHO databases. One possible explanation of differences in deaths rates may however be in the systematic way in which the data are collected. Are there any know similarities or differences in the way in which each jurisdiction collects and codes cancer causes of death? A recognition of this being a possible limitation is necessary.
Tabulated data are provided in the supplementary material. Data are available from WHO.
Conclusions relating to the overall trends showing increasing mortality rates are generally appropriate. These results can and should be used to direct initiatives to inform public health programs and campaigns. The relationship between UV index and mortality is speculative and requires more robust and detailed analysis.

Additional comments

This paper is an interesting addition to reports on mortality from melanoma. The 4 populations chosen somewhat limit the ability to generalise the findings to the wider geographical area. The two most interesting issues are the increasing trend over time and the difference in rates between males and females. The authors put these in the context of international trends.
My general comments relating to issues raised above:
1. The English requires some editing. There are sections of extraneous material that could be edited to make the paper brief and to the point.
2. The presentation of the relative effects on age standardised mortality rates is little unusual but appears useful in comparing rates between areas and between gender.
3. The correlations between UV levels and mortality is a very crude ecological approach.

---

## Round 0.2 · Minor Revisions

· Academic Editor

Minor Revisions

I thank you for responding to the previous comments as you have done. There are some editorial changes that need to be incorporated into the manuscript before it can be accepted for publication. These changes have been detailed in this response.

Reviewer 1 ·

Basic reporting

It is generally fine.

Experimental design

The revised version was imporved. The data of mainland China are available. Authors could find those from Cancer Registry Annual Report (last volumn is the year of 2012). It could be better to use data covered more areas and populations. However, it is accepted using recent data.

Validity of the findings

It is ok now.

Additional comments

The revised manuscript is better and comments from reviewers have been responsed appropriately.

Reviewer 2 ·

Basic reporting

The basic reporting is satisfactory.

Experimental design

Methods and design are appropriate and well described.

Validity of the findings

Findings have been clearly articulated and the discussion is appropriate and balanced.

Additional comments

Thank you for the opportunity to undertake a second review of this manuscript. The authors have responded appropriately to the comments and suggestions I made in the initial review and I recommend PeerJ accept this manuscript. The authors should be congratulated for an insightful and important piece of work.

---

## Round 0.3 · accepted · Accept

· Academic Editor

Accept

Hi - thanks for making those changes. I think the paper is much clearer as a result, and I appreciate your willingness to make these revisions. I have asked the PeerJ staff to consider revising Figure 1, so that the sex, age group and region effects are on the same scale.